# Review of the Organizational Structures of the Trail Running, Skyrunning and Mountain Running Modalities in Spain

Sergio López-García [1], Jaime Muriel-Isidro [1], Brais Ruibal-Lista [2,*], Rubén Maneiro [1] and Mario Amatria-Jiménez [1]

1    Faculty of Education, Pontifical University of Salamanca, 37007 Salamanca, Spain
2    EUM Fray Luis de León, Catholic University of Ávila, 47010 Ávila, Spain
*    Correspondence: brais.ruibal@frayluis.com

**Abstract:** The purpose of this research was to analyze the organizational structures of the different federations and entities representing the modalities of trail-running, skyrunning, and mountain running and how they have settled in the current sports landscape. The main task of these entities is to act as hosts of the legally established modalities, both internationally and nationally, applying their respective regulations and statutes unilaterally. Through an in-depth review of the different regulations, statutes, and current articles of all the organizations that represent the different disciplines, we can observe how a sport modality that seems to be the same is nevertheless distant in its execution (regulations, championships, classifications, systems of points, etc.), which has led to the international level being established in different sport modalities (trail running and skyrunning). At the level of the Spanish territory, it has led to an administrative dispute to know who has the powers to approve sports competitions.

**Keywords:** trail running; skyrunning; mountain running; modality structure; administrative bilaterality

## 1. Introduction

In the last two decades, sports modalities carried out in the natural environment have appeared and proliferated as emerging disciplines associated with risk, tourism, adventure, and competition [1–3].

The proliferation and emergence of new sport modalities over the last 10 years has been remarkable. At an international level there is a constant renewal of the sports catalogue, and in Spain this situation has not been any different. A clear example is trail running (TR), skyrunning (SR), and mountain running (MR), which have settled in the sports scenario, especially in Spain, where they are highly accepted among runners.

Mountain running (MR) is a sport practice that in recent years has been studied and analyzed, taking into account the social, media, environmental, and economic impacts that it has and is generating in society and the Spanish region [4].

This new modality, besides growing in importance in the sports field, has also generated relevant importance in others, such as tourism, environment, safety, and equality, and has also generated at the same time the emergence of several bodies in charge of managing it, such as federations, institutions, clubs, and companies. According to Seguí et al. [5], this growth has led to "a situation where the struggle to achieve recognition of the sport is ongoing", both on an international and a national level.

This struggle to recognize who should organize and sanction the sport in all its different forms in Spain has given rise to a series of administrative problems, leading to a succession of resources and appeals between the Spanish Federation of Mountain and Climbing Sports (FEDME) and the Royal Spanish Athletics Federation (RFEA) from 2018 to the present, leaving this decision in the hands of the National Sports Council [6] (CSD, for its Spanish initials—Consejo Superior de Deportes), the General Sports Directorates of each Autonomous Community, and the provincial courts.

As a result of the administrative problem, chaos has arisen regarding the way in which each entity competing for the management and administration of the modality has built its organizational structure. They have programmed their competitions and regulations unilaterally as well as their vision of the modality, which has brought about a series of inconveniences to runners and direct organizers of the championships when competing and certifying the courses of the competitions of each event, which are governed, among others, by the Law of Public Shows of the Autonomous Communities [7,8] (Ley de Espectáculos Públicos de las Comunidades Autónomas, CC.AA.). This law decides, according to the Autonomous Community where the event is held, who will be responsible for the organization of the event. The purpose of this research was to analyze the organizational structures of the different federations and entities representing the modalities of trail running, skyrunning, and mountain running and how they have settled in the current sports landscape.

## 2. Modality Definition

Before mentioning how the different organizations have defined the modality they manage, it is important to emphasize what running in the natural environment means. A distinction must be made between the regulated and non-regulated practice of what we understand by running in mountain areas, which according to the terminology of the Regulations on Natural Areas II of the Spanish Federation of Mountain Sports and Climbing [8], is as follows: "The modality consists of running through mountain areas".

This modality involves moving through different natural environments with highly variable morphological and meteorological characteristics, which often involves considerable physical effort [9,10].

Even if it is with only one organization, we are aware of the implications of running in mountain areas, but what are we talking about when we refer to TR, SR, and MR? In order to approach it, we will explain how the different organizations define them both at a national and international level. The conclusion drawn from this is that they always refer to the same sport modality.

The International Association of Athletic Federations (IAAF) defined trail running in article 252 of the competition regulations as follows: Trail runs take place on a wide variety of terrain (including dirt roads, forest roads, and single-track trails) in a natural, open-field environment (such as mountains, deserts, forests, or plains) and mostly off-road. Paved or concrete-surface pathways are acceptable, but they must be the minimum possible to achieve the desired run and not exceed 20% of the total run distance. There is no limit for distance or elevation gain uphill or downhill [11]. It is managed by its associate members, such as the International Association of Ultrarunners (IAU) and the International Trail Running Association (ITRA), who organize (in the case of the IAU and those sponsored by the IAAF), in addition to their own competitions, the TR World Championship on a jointly basis.

Article 251 of the competition regulations includes mountain running (MR), which the IAAF [11] has defined it as follows: "Mountain running is carried out on terrain that is mostly off-road, unless there is a significant elevation change, in which case a paved surface is acceptable. Distance can vary from 1 km to marathon length. The slope should be at least 5% and should not exceed 20%". It is managed at the international level by the World Mountain Running Association (WMRA), an associate member of the IAAF, which also defined mountain running (MR).

In addition, article 250 of the competition regulations also recognizes cross country, which has been defined as "races through open or wooded areas on grassy terrain, which may include partly gravel, roads, and hills as long as these are limited to a minimum and also include natural obstacles" [11].

The ITRA [12], an associate member of the IAAF, presented TR as "a foot race, open to all, in a natural environment (mountain, forest, plains), with the minimum possible number of asphalt or paved routes, which should not exceed 20% of the total course".

On the other hand, the American Trail Running Association [13] made a broader description of the concept of trail runs, which was defined as "races that not only run on off-road trails but can also include paved roads in rural and urban areas," while adding that in order to be considered a trail run, they must run on tracks or roads inaccessible to motor vehicles.

The Ultra-Trail® World Tour [14], a private race circuit, made it clear that trail runs should illustrate the diversity of the event, including steep roads, trails, hills, beaches, or desert, by applying for the definition of the Ultra-Trail® circuit, which includes five unique criteria that the event must have, such as a distance of at least 100 km, an emblematic location, at least 500 participants, an international event with at least 20 different nations represented, and at least two editions of the event having been held.

It should be noted that Ultra-Trail® is a registered trademark of the company organizing the Ultra-Trail du Mont Blanc® (UTMB), more specifically by the SARL Autour Du Mont Blanc, directed by Catherine Politte and Michel Politte, and that has co-organized, in addition to the UTMB, the ByUTMB franchised racing circuit around the world since 2018 and the UTWT, to which the UTMB belongs (they are one of the eight most important event organizers in the world, being the founding members of the UTWT in 2013). Moreover, they are both founding members of ITRA, as stated in the statutes of 2 February 2019, approved in Milan [12].

On the other hand, we can mention the International Skyrunning Federation (ISF), which represents the governing authority on SkyRunning® and mountain running above 2000 m carried out on technical trails and whose practice is mainly included in the Sky, Ultra, and Vertical categories [15]. According to the statutes of 23 July 2011 and the regulations of the competition, the definition for SR is as follows: "Running in the mountains above 2000 m of altitude, where the difficulty to climb does not exceed degree II and the slope is greater than 30%. Where the altitude does not reach 2000 m, the courses must have an average minimum of 6% of slope over the total distance and reach the highest points of the area".

In the case of SkyRunning® it should also be remembered that it is a patent registered by Marino Giacometti (inventor of SR and founder of the ISF) and that it is exploited through SkyMan S.A.

In Spain we can find several organizations that have defined the modality practically like their international counterparts, with the differentiation of concepts as far as the name is concerned. Thus, the Royal Spanish Athletics Federation [16], under the protection of the IAAF regulations, is in charge of managing trail running, MR, and cross country in Spain, defining TR as:

> "Activity that consists of running in a natural environment in the open field (mountains, deserts, forests, beaches, or plains) where no more than 20% of the surface can be asphalted or paved. The path can be diverse (roads, trails, tracks . . . ) and the route must be properly marked. Trail running allows a great variety of distances, slopes, terrains, and landscapes".

The Spanish Association of Trail Running (AET) uses the same explanation given by the ITRA to define TR and in recent years has taken the same steps as its international counterpart (Table 1).

**Table 1.** Organizational structure of the trail-running and skyrunning modalities at an international level.

| Organization | Modality | Championships | Categories | Ranking | Collaboration with |
|---|---|---|---|---|---|
| ITRA | Trail running | Trail Running World Championship (co-organizer) | Individual and teams | According to ITRA Performance Index | IAU and UTWT |
| IAU | Ultra-distance | Trail Running World Championship (co-organizer) 24 h, 100 km, and 50 km World Championships | Individual and teams | Annual ranking | IAAF and ITRA |
| UTWT | Ultra-Trail® | Series Bonus, Series, Pro, Challenger, and Discovery Races | Individual | World ranking and annual ranking | ITRA and UTMB |
| IAAF | Cross-country | Cross Country World Championship | Sub-20 and individual | According to race position | IAU, WMRA and ITRA |
| WMRA | Mountain running | CDM World Championship, Long Distance CDM World Championship, Veteran CDM World Championship, CDM World Cup, and CDM Under-18 International Cup | Individual, teams, and national teams | Annual ranking (last 12 months) | IAAF |
| UTMB | Ultra-Trail® Trail running | El UTMB©, Courmayeur-Champex-Chamoix (CCC®), Sur les Traces des Ducs de Savoie (TDS®), Orsières-Champex-Chamonix (OCC), Martirgny-Combe à Chamonix (MCC), Petite Trotte à Léon (PTL®), and Youth Chamonix Cormayeur (YCC) | Individual by age | According to race position | ITRA and UTWT |
| ISF | Sky, Ultra, Vertical, Skyspeed, Vertical Running, Skybike, Skyraid and Skyrunning Technical Level | Skyrunning National Championships, Continental Skyrunning Championship, Biennial World Skyrunning Championships, Skyrunning Junior World Championships, Skygames, World Vertical Circuit, Vertical Kilometre, World Circuit Skyrunner National Series, and Skyrunner World Series | Individual, teams, national teams, U23, and combined | According to position in certain races and general ranking by series | - |
| GTS | Trail | Golden Trail World Series (GTWS), Golden Trail National Series (GTNS), Golden Trail Championship (GTCS) | Elite, individual, and teams | General ranking | - |

ITRA: International Trail Running Association. IAU: International Association of Ultrarunners. UTWT: Ultra Trail World Tour. IAAF: International Association of Athletics Federations. WMRA: World Mountain Running Association. UTMB: Ultra-Trail of Mont-Blanc. ISF: International Skyrunning Federation. GTS: Golden Trail Series.

Finally, at the national level, the other major protagonist of the mountain races in Spain is the Spanish Federation of Mountain Sports and Climbing [17], affiliated with the ISF and which has defined them as follows: "Mountain races consist of running along paths or virgin areas that are characterized by their steep slopes and their technical difficulty. They can be held in the high, medium, and low mountains, and always run along unpaved tracks and courses, paths, ravines, etc. and the race route does not exceed 50% of the track accessible to vehicles. The minimum distance for a circuit to be considered an official competition, except in the case of the Vertical Kilometer, is 21 km, with a minimum accumulated difference in height of 1000 m. Depending on the terrain on which the competition takes place, the races may include climbing up and down with ropes, although the area climbed may not exceed grade II of difficulty or 40° of slope".

This definition of MR is practically identical to that of SR, with the difference in the conceptualization of what both organizations understand by their respective ways of naming the modality. Although at the international level the ISF has wanted to make a distinction between its modality as one of its own, SR, against another that is considered different at the level of the Spanish territory—TR—this has remained an administrative dispute over the term "TR," to which both the RFEA and the FEDME refer as their own, both understanding that TR is either "Carrera de montaña" (RFEA) or "Carrera por montaña" (FEDME), both called in English "mountain running".

### 3. Administrative Bilaterality and Consequent Problems

Although apparently the modality lies in a similar sporting situation consisting of "running in a natural environment", the disputes over its management and all that it entails are being patented in Spain, especially in recent years in which there has been an explosion in the number of runners and organizers, identified as the trail-running boom according to Zagalaz et al. [18] (as quoted in [4]). Although at the international level the struggle in the sport has been mainly in the dispute regarding certain of its disciplines, such as the Ultra-Trail®and Ultra SkyMarathon®(which would correspond to ultra-distance races), the ISF has opted to abandon the fight for the longer ultras, a field in which the ITRA (member of the IAAF) is better positioned alongside the UTWT, ultimately opting for shorter races of between 22 and 66 km with a marked technical character that returns it to the origins of SR, differentiating it from TR as a separate sport [19].

Although at an international level it seems that the modality has been divided into two different sports (TR and SR), in Spain, the situation is more complicated, with the dispute over the competitions of the modality and the authorship of the term Trail Running® (which we must remember is a registered trademark of the owners of the UTMB®) persists. On the one hand, in 2013 FEDME presented an amendment to article 3 of its statutes (which includes the sport modalities that this federation manages, including MR) to the CSD (National Sports Council), which was recognized in the Spanish Official Gazette (BOE—for its Spanish initials—No. 270, Resolution 4 November 2013) [6] and which granted it the powers of the mountain-running (MR) modality, also commonly known as TR in Spain.

The conflict lies in the fact that the RFEA, in view of the trail running boom and as a promoter of MR—established in its 2014–2015 regulations as the only discipline carried out (within its organization and apart from cross country), until that same year, off-road or on a paved surface with a specific profile (average slope of 5% and maximum of no more than 20%)—decided to include TR as an athletic discipline following the steps of the IAAF, which recognized it at the International Congress in Beijing in August 2015. Thus, until 2015, only the following types of MR could be found in the competition regulations of this federation: (a) classic mountain races (with no more than 12 km of distance and a maximum elevation gain of 1200 m uphill), (b) long-distance races (these cover distances between 20 and 42.195 km approximately and an elevation gain of a maximum of 4000 m, with those under 18 years old not being able to compete in distances greater than 25 km), (c) mountain relay races, and (d) mountain time-trial races [20].

The total conflict for the exclusivity of the term TR and said modality in Spain comes with the Resolution of 26 September 2018 by the National Sports Council (CSD, for its Spanish initials) presidency, which recognizes an amendment to the RFEA statutes in articles 1, 16, 20, 23, 60, 64, and 73 (Spanish Official Gazette—BOE No. 240, Resolution 26, September 2018) [21]. In article No. 1 of this document, TR is recognized as an athletic discipline belonging to the RFEA, whose valuable basis is the definition that the international organizations (ISF and IAAF), with which both federations are affiliated, provide of the disciplines of SR and TR, respectively [22].

After this resolution of the CSD in 2018, the appeals of the FEDME and several of its autonomous federations before the ordinary justice followed one another (Recourse National Court 19/2020 [23], with sentence on 19 February 2020, and Appeal National Court 27/2020 [24], with sentence on 17 July 2020). These resolutions ruled in favor of the RFEA, being dismissed and allowing the execution of both modalities as different. On the one hand, the competencies of TR were established as a modality belonging to the RFEA and, on the other hand, the CPM, whose competencies would remain in the hands of the FEDME.

Ultimately, the Central Administrative Court No. 5 of Madrid issued a ruling in March 2022 (Sentence No. 61/2022) [25] upholding the appeals filed by FEDME together with the Basque, Navarran, and Aragonese mountain and climbing-sport federations. Andalusia, against the CSD resolution of 26 December 2018 [26], annulled it.

With this resolution, the current legal insecurity in the field of management of the modality in Spain was evident, with provisions of the Law on Public Shows and Recreational Activities (of that same Autonomous Community; see Law 3/2017 of 5 April [7] on Public Shows and Recreational Activities of Cantabria in Article 7 Section f, or Law 7/2006 of 2 October on Public Shows and Recreational Activities of the Community of Castile and León [8], Article 14 Section 2a), and the general sports delegations being the ones responsible for determining which organism or federation is in charge of authorizing the organizers of TR or SR events according to the definitions provided for their respective disciplines and of the probative sentences that both federations have contributed in the different judicial processes (Table 2).

**Table 2.** Organizational structure of the trail-running and mountain-running modalities in Spain.

| Organization | Modality | Championships | Categories | Ranking | Collaboration with |
|---|---|---|---|---|---|
| RFEA | Trail running Mountain running (international) | Spanish Mountain Racing-Trail Running Championship, Spanish Individual Trail Running Championship, Spanish Master Mountain Championship, Spanish Master Trail Running Championship, Spanish Trail Running Federation Championship, Spanish Trail Running Club Championship, Spanish Mountain Racing Club Championship, and Cross Country World Championship | Individual, Veteran, Clubs and Federations | According to race position | IAAF |
| FEDME | Mountain running (national) | Online Mountain Racing Spanish Cup, Online Mountain Racing Spanish Championship, Vertical Mountain Racing Spanish Cup, Vertical Mountain Racing Spanish Championship, Ultras Spanish Championship, Ultras Spanish Cup, and Spanish Snow Mountain Racing Championship | Individual, Clubs and Federations | Ranking according to discipline (line, vertical, or ultra), type of competition (cup or championship), position (1 to 15), and gender | ISF |

RFEA: Real Federación Española de Atletismo. FEDME: Federación Española de Deportes de Montaña y Escalada. IAAF: International Association of Athletics Federations. ISF: International Skyrunning Federation.

From this situation of administrative instability of the modality, several problems have arisen that affect the athletes and organizers of the events. Among them, the following stand out:

(a) Athletes and technicians:

- Planning of seasons and training: Nowadays it is chaotic to make a calendar in optimal conditions and in many situations; how to organize a valid training session for different types of races is a dilemma.
- Access to competitions and classifications: Certain competitions demand a number of specific points from that same competition (UTMB or SkyMasters) to be able to participate, which must be obtained in other races approved by third parties who own the ranking or classification (UTMB, UTWT, ITRA, ISF, RFEA, FEDME), such as points from ITRA, UTMB, UTWT, ISF, or their circuits with their own classifications and rankings. This can mean that by participating in one race, one may not score points to enter another or may not manage to score points in a certain classification, with both cases being the main objectives of the athlete, whether amateur or professional.
- Safety: Different trials require different material and human resources, as well as other risk-management systems.
- Insurance, licenses, and coverage: Athletes often participate in different competitions with specific insurance or licenses for each one, which leads them to being federated and insured with multiple federations and organizations.

(b) Event organizers:
- Permits: A conflict of interest may exist in the Autonomous Communities where it is determined that a federation must be the one to authorize the competition.
- Certified courses: Participating in a championship or organizing an event of this type means certifying the race according to the guidelines of the corresponding federation on the subject. In turn, this means adapting the regulations, safety systems, anti-doping protocols, and others, which in most cases entails high fees for the organizer in addition to a greater number of administrative requirements.
- Structural changes in the event races: Significant changes must be made in route, material, human, and technical resources in order to comply with the standards, which means, once again, greater economic cost and fewer benefits.

## 4. Organizational Structure of the Modality

Sports regulated by sports federations and entities establish the management and administration of the sport on the basis of legal regulations, their respective rules, and their statutes. In the case of TR and SR, since they are sports that have evolved from an amateur perspective to a highly competitive one while conserving an added popular value with a particular meaning as far as terminology is concerned and taking into account that there is a good number of organizations in charge of their management (both at the international and national levels), it is easy to see that there is not one single classification system for them. Seguí et al. [5] pointed out that in relation to the organizational structures of mountain and trail running, three basic aspects are considered: (a) the organizations carrying out the activity of mountain and trail running, (b) how they define the sport modality, and (c) how they conduct their respective championships.

From these three aspects listed by these authors we already know the organizations and their respective definitions. As far as the organization of their championships is concerned, we must also add the subdisciplines of the modality, the different types of races, the categories, and the scoring and classification systems. Therefore, on a worldwide level, the following can be found:

(A) The ITRA, as an international promoter of TR, has organized together with the IAU the World Championship of Trail Running (Individual and Team categories) since 2016. With the amendment by the IAAF of article 252 of the competition regulations, the ITRA decided in March 2018 to implement a new classification system for its races from XXS to XXL, based on the km/effort that the runner must exert, validating a new system of ITRA points (understood as a performance index) ranging from 0 to 6 depending on the km/effort. They have also collaborated in the organization and regulation of the Ultra-Trail® race circuit of the UTWT.

(B) The IAAF, in addition to the trail races described in article 252 of its competition regulations and that it shares with the ITRA, has among its disciplines cross country, which was ruled in article 250 of its regulations; mountain running (MR), which was delegated to its associate member WMRA and whose regulation is located in articles 251 and 252; and ultra running, which was also delegated to the IAU for its management and organization of the respective championships sponsored by it and which was regulated in the same way as trail races and MR in article 252. Consequently, only the World Cross Country Championships are organized directly by the IAU, whereas the other disciplines are organized by its associate members.

(C) The WMRA is the promoter of mountain racing worldwide. It organizes basically five events: World Mountain Racing Championship (Individual and Team categories), World Long Distance Mountain Racing Championship (Individual and National Team categories), World Masters Mountain Racing Championship, World Mountain Racing Cup (Individual category), and International U-18 Mountain Racing Cup (Individual and Team categories). It has a non-stop ranking in which the results obtained in the last 12 months are scored.

(D) The IAU is responsible for promoting ultra-distance races (not only trail races but also asphalt, indoor, and track races). It has worked together with the ITRA since 2016 to organize the World Trail Running Championship. In addition, it organizes the 24 h, 100 km, and 50 km World Championships. Among its disciplines, as stated in its articles of incorporation of June 2016 (revision), are 50 km, 6 h, 100 km, 100 miles, 24 h, 48 h, 6 days, and 1000 miles for road and trail. It consists of a unique annual ranking for the disciplines of 50 km, 100 km, 100 miles, 6 h, and 6 days, in which one scores by completing races whose titles are Gold, Silver, and Bronze (with Gold granting more points and Bronze fewer).

(E) The UTMB® has organized a unique annual event made up of seven races according to its competition regulations, which are the UTMB©, the Courmayeur-Champex-Chamoix (CCC®), the Sur les Traces des Ducs de Savoie (TDS®), the Orsières-Champex-Chamonix (OCC), the Martirgny-Combe à Chamonix (MCC), the Petite Trotte à Léon (PTL®), and finally, the Youth Chamonix Cormayeur (YCC). For participation in the first four races, it is necessary to accumulate a certain number of UTMB points, which can be obtained in third-party races authorized by the UTMB® called "qualifying races", in which one can obtain from one to six UTMB points.

(F) The UTWT is another major protagonist in Ultra-Trail® races due to being the main promoter of this distance in collaboration with the ITRA. In its competition regulations it presents an annual circuit of races in different categories and similar standards established jointly with the ITRA. Among the racing categories are Series Bonus, Series, Pro, Challenger, and Discovery races (the latter proposed as demonstration races with no possibility of scoring for the circuit). The UTWT currently has two classifications: the UTWT Annual Ranking, which takes into account the two best results obtained in races on the circuit as long as at least two of them are completed, and the UTWT World Ranking, which takes into consideration the five best results obtained in the last three years. In any case to qualify for any of the two rankings a series of UTWT points must be obtained, which depends on the category of the race and the position that each runner obtains.

(G) The ISF shows a relatively more complex structure as far as the organization of its modality is concerned. The following disciplines were included in the competition regulations for the year 2019 [27]: Sky, Ultra, Vertical, Skyspeed, Skyscraper Racing/Vertical Running, Skybike, Skyraid, and Skyrunning Technical Level. Furthermore, the following terms are used to describe the types of races they offer: SkyRace®, SkyMarathon, Ultra SkyMarathon, and Vertical Kilometer®. Meanwhile, it has approved the following competitions: National Skyrunning Championships (developed by each nation based on the ISF regulations that have, firstly, an individual scoring system and secondly, a qualifying system based on the four best athletes in order to make up the national team, at least one per gender), Continental Skyrunning Championship (individual and team competition that has at least the categories of Sky, Vertical, and Ultra), Biannual World Skyrunning Championship (at least the Sky, Vertical, and Ultra disciplines are competed for and the title of each one is awarded not only combined, but also for national teams), World Youth Skyrunning Championships (held in the disciplines of Sky and Vertical, as well as three different categories—A, B, and U-23, which are awarded in the two disciplines), and Skygames (this type of competition is held every four years, coinciding with the Olympic Games in the disciplines of Sky, Vertical, and Ultra, but may also include others, such as Skyspeed, Skybike, or Skyraid, establishing an individual title for each of the disciplines as well as a combined title and another one for teams). In addition to the official championships, the ISF has also arranged a series of circuits with third parties that maintains the Skyrunner® and Vertical Kilometer® trademarks, where individual runners and teams can participate. Their importance depends on the media impact of these circuits. These circuits are the Vertical Kilometer World Circuit (races developed in the most emblematic skyscrapers in the world with only one general ranking of individual character), the Vertical Kilo-

meter World Circuit (vertical kilometer racing circuit with only one general ranking of individual character), the Skyrunner National Series (with two absolute categories, male and female, in the Sky and Vertical disciplines and with a qualification system awarding points to the top 30; within these races, only those within the SkyMarathon modality score for the Skyrunner World Series and only the top 10 in each category), and the Skyrunner World Series (races with a single category called SkyRace, in which it is only possible to participate individually and whose scoring system is based on the name of each race, which may be SkyRace, SuperSky Races, or The SkyMasters. Each denomination offers a certain number of points according to the athlete's position, taking into account only the first 20 of each category) [28].

(H) Finally is the Golden Trail Series (GTS), with its two circuits—the Golden Trail World Series (GTWS) and the Golden Trail National Series (GTNS)—with unique absolute categories for each type and a classification system awarding points to the top 30 in each race.

Meanwhile, at the national level, the institutions with their own organizational structures and with real capacity to manage the TR/SR modality are:

(A) The RFEA [20], acting in accordance with the IAAF, approved in its competition regulations the following disciplines at the Spanish level: cross country, and mountain and trail races, and on the basis of these disciplines, the following competitions have been organized: the Spanish Championship of Mountain Races—Trail Running, the Spanish Individual Championship of Trail Running, the Spanish Master Mountain Championship, the Spanish Master Trail Running Championship, the Spanish Championship by Trail Running Federations, the Spanish Championship of Trail Running Clubs, and the Spanish Championship of Mountain Running Clubs. In addition to those mentioned, the World Cross Country Championships are also organized in the Individual, Masters, Clubs, and Federations categories.

(B) Additionally, FEDME described the following mountain-race disciplines in its 2019 competition regulations: Line Races, Vertical Races, Ultra Races, Mountain Snow Races, and other state-wide competitions and popular races. The different competitions authorized by FEDME are specified as follows: the Spanish Cup of Line Mountain Races (it consists of between three to six races of one day each and they are disputed in the individual modality), the Spanish Championship of Line Mountain Races (it consists of a single race in the individual modalities by clubs and by autonomous federations), the Spanish Cup of Vertical Mountain Races (it consists of between three to six races in the individual modality), the Spanish Championship of Vertical Mountain Races on the Mountain (it consists of a single race in the individual modalities by clubs and autonomous federations), the Spanish Ultras Championship (it consists of a single event in the individual modalities by clubs and autonomous federations of a maximum of two days), the Spanish Ultras Cup (it consists of between three and six events in the individual modalities of a maximum of two days each), and the Spanish Championship of Snow Mountain Races (it consists of a single event in the individual modalities by clubs and autonomous federations). In addition to the disciplines and competitions, FEDME has its own ranking, which takes into account the different disciplines (line, vertical, or ultra) according to the competition (cup, championship, or certified course), rating the latter differently and always taking into account those classified between positions 1 and 15, dividing in any case the ranking by male and female.

## 5. Conclusions

From this administrative chaos, it can be inferred that what at first sight was presented as a single sport modality that projected a similar image in all competitions, whose common factors are running and the natural environment in which they are carried out, two different disciplines are struggling for a similar modality in view of future participation in the Olympic Games, with the ITRA fighting for it together with the IAAF on the one hand, and

on the other hand the ISF together with the International Union of Alpine Associations (UIAA), which could lead to TR and/or SR becoming Olympic disciplines in the next 10 years as of Tokyo 2020.

Therefore, it is important to remember the following:

(1)  Although TR and SR may be considered the same modality, they imply different sports preparations both at a competitive level (athletes and trainers) and even within their own sub-disciplines (taking into consideration that planning and training methods vary according to each), as well as at an organizational level (competition organizers), which implies different adaptations of regulations, certifications, and institutional permissions, always depending on the sports federation to which the competition belongs.

(2)  At the international level, the corresponding federations and institutions already defend their modalities as something separate: Whereas the ITRA and IAAF defend trail running as their own, the ISF has differentiated skyrunning as a modality of its own apart from TR, being much more technical and specific, which the best runner to be judged in all fields (Sky, Vertical, Ultra, etc.), thus marking a separation from TR.

In Spain the situation is still complicated and unspecific, because despite the latest rulings by the CSD, the National High Court, and the appeals filed by the different Autonomous Mountain Federations before the Central Contentious-Administrative Court No. 5 of Madrid, both federations continue to battle for hegemony over the modality. Although the RFEA (2022) has made it clear that it contemplates continuing with the management of the TR modality in all its facets (everything that is running in the natural environment and is not understood as CDM or cross country), FEDME, which has been reinforced with the latest favorable ruling (Sentence no. 61/2022), continues to defend CPM as its modality, arguing all the issues related to the legal system of sports modalities in Spain and referring to the complete definition that the ISF (2019) made in the preamble to its statutes on the SR [29], in which, in addition to the definition itself, it warned that "*when the altitude does not reach 2000 m, the courses must have a minimum average inclination of 6% over the total distance and reach the highest points of the area*". In any case there are several possible solutions to the problem: (a) to determine TR and the SR as two different sport modalities in the same way that it has been done at a worldwide level; (b) to grant the modality legally (it should be remembered that both FEDME since 2013 and RFEA since 2018 are legal in this aspect, as stated in their statutes and recognized by the National Sports Council or CSD) to only one federation, as has happened in other more powerful countries in this field, such as France or the United States, where it is the Athletics Federation that manages and administers the modality; or (c) the last and most improbable one, which consists of formalizing a separate trail-running federation to independently exploit the modality, which is very difficult considering that following the same line explained by Seguí et al. [2], the institutions that could fight for it, such as the AET, "should mutate its legal form to that of a federation in order to be able to compete, in the future, with the RFEA and the FEDME".

**Author Contributions:** Conceptualization: S.L.-G., J.M.-I. and R.M; investigation, S.L.-G., J.M.-I. and B.R.-L.; writing—original draft preparation: J.M.-I., B.R.-L. and R.M.; writing—review: S.L.-G., R.M. and M.A.-J.; visualization: S.L.-G., J.M.-I., B.R.-L. and R.M.; supervision: S.L.-G., J.M.-I., B.R.-L., R.M. and M.A.-J. All authors have read and agreed to the published version of the manuscript.

**Funding:** This research received no external funding.

**Institutional Review Board Statement:** Not applicable.

**Informed Consent Statement:** Not applicable.

**Acknowledgments:** Thanks to Cristina Muriel Isidro, Spanish teacher at the Institut Gaspard Monge (Sagvigny sur Orge, France), for her collaboration and advice in terms of translation.

**Conflicts of Interest:** The authors declare no conflict of interest.

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
