# Peer review of "Review of the Organizational Structures of the Trail Running, Skyrunning and Mountain Running Modalities in Spain"

_sustainability, doi:10.3390/su141912401_

Round 1

Reviewer 1 Report

The manuscript titled "Trail-Running, Skyrunning and Mountain Running: Organizational Structures of Modalities" presents a review of the problems that exist in Spain regarding the organization of these sports modalities.

It is an interesting article, although it is considered that aspects should be improved for possible publication in this journal:

-       Given that the review deals with a reality that only affects Spain (no data is provided that the same thing happens in other countries), it is recommended to modify the title of the work. For example: “Review of the organizational structures of the trail running, skyrunning and mountain running modalities in Spain”

-       As previously stated, the article focuses on the specific situation in Spain, so this situation must be clarified or nuanced throughout the manuscript.

-       Line 42 refers to the CSD (National Sports Council), however line 176 indicates it as CDS. Correct.

-       It is necessary to clarify more precisely the organizational responsibility of each of the tests, since it is not clear who has the competence in each case throughout the manuscript.

It is considered that these recommendations, once applied, will improve the quality of the manuscript so that it can later be published by the journal.

Author Response

Dear reviewer, nice to greet you.

First of all, I want to thank him, on behalf of all the authors, for the extensive review he has done of our work.

I will list the changes applied:

  1. We have changed the title. We have selected the one you indicated in the review.
  2. We have clarified that this work is set in Spain. You can see the directions on lines 31, 33, 44, 51, 141, 144, 155, 176, 187, 195, 208, 229, 432 and 440.
  3. We have corrected the CSD acronym throughout the document.
  4. Lastly, we have prepared two tables where we indicate those responsible for the different tests both at the international level (Table 1) and at the state level in Spain (Table 2).

Finally, we would like to point out that we have also added new legislative information that has been published in the last 2 years and that we consider relevant. You can consult it on lines 216 to 227 and 432 to 444.

We hope that these modifications are enough to improve the quality of the work and that it can be published in Sustainability.

Receive a cordial greeting from all the authors.

Reviewer 2 Report

Dear authors

Exercise is beneficial to promote human health. The right promotion of sports organizations can help promote people's health benefits and achieve health recovery results. It's an interesting and challenging topic to discuss what the "right organization" is. However, the discussion in the text needs a more fair and clear discussion, so the following suggestions are proposed for revision.
1. Summary
Although it is a "Review" mode, the basis for this analysis data still needs to be supplemented. For example, collect 2000-2022 related literature...
2.Introduction
"As a result of the administrative problem....", please clearly indicate what kind of confusion, and provide the basis.
3. Modality definition
Is "Ultra-Trail®" "Ultramarathon" ? .
In this chapter, the author proposes a lot of literature for comparison, but it is difficult to compare the textual layout and description. In my opinion, by making a table for analysis, it is easier for readers to understand the definitions of Ultra-Trail-related sports and competitions, hosting goals and responsibilities of different associations and units.
4. Organizational structure of the modality
As noted above, the data, although categorized sequentially, explain the goals of the work of each organization. But it is not easy for readers to make judgments.
5.Conclusions
Although the conclusion has a certain meaning, it is not easy to obtain results based on the knockdown process.

Finally, although the manuscript is in ""Review" mode, I don't think that articles can only be presented in words. Presenting in tables or pictures at the right time should highlight the meaning of this type of theme.
I will be expecting substantial improvements from the author.

Author Response

(The authors gave the same response as above.)

Reviewer 3 Report

This research sought to examine some federations and entities that represent the modalities of trail running, sky running, and mountain running, their organizational structures, and how they have adapted to the contemporary sports environment.

The study is well written with good rationalization and make valuable conclusion related to distinguishing these sports. I suggest authors look at the manuscript for language check before publication.

I congratulate authors for this well prepared work. 

Author Response

Dear reviewer, nice to greet you.

First of all, I want to thank him, on behalf of all the authors, for the extensive review he has done of our work.

We appreciate his words as this is not a "normal" research study, as it involves reviewing many administrative court rulings and trying to put all available information on the table.

Finally, we would like to point out that we have also added new legislative information that has been published in the last 2 years and that we consider relevant. You can consult it on lines 216 to 227 and 432 to 444.

Receive a cordial greeting from all the authors.

Round 2

Reviewer 1 Report

Thank you very much for taking my considerations into account.

Reviewer 2 Report

Dear author
Glad to see the revised manuscript. I think the current situation is enough to improve the visibility of the manuscript and readers will be able to gain clear, easy-to-read knowledge.
Therefore, I suggest that the editor-in-chief may consider adopting this manuscript for publication in a journal.